# A Closed-Loop Ear-Worn Wearable EEG System with Real-Time Passive Electrode Skin Impedance Measurement for Early Autism Detection [note 1]

**DOI:** 10.3390/s24237489

**Published:** 2024-11-24

**Authors:** Muhammad Sheeraz, Abdul Rehman Aslam, Emmanuel Mic Drakakis, Hadi Heidari, Muhammad Awais Bin Altaf, Wala Saadeh

**Affiliations:** 1Department of Bioengineering, Imperial College London, London SW7 2AZ, UK; m.sheeraz@imperial.ac.uk (M.S.); e.drakakis@imperial.ac.uk (E.M.D.); 2Department of Computer Engineering, University of Engineering and Technology Taxila, Taxila 47050, Pakistan; 3School of Engineering, University of Glasgow Scotland, Glasgow G12 8QQ, UK; hadi.heidari@glasgow.ac.uk; 4Engineering and Design Department, Western Washington University, Bellingham, WA 98225, USA; awaisbinaltaf@gmail.com (M.A.B.A.); saadehw@wwu.edu (W.S.)

**Keywords:** autism spectrum disorder (ASD), electroencephalogram (EEG), chronic neurological disorder (CND), electrode–skin impedance (ESI), analog front end

## Abstract

Autism spectrum disorder (ASD) is a chronic neurological disorder with the severity directly linked to the diagnosis age. The severity can be reduced if diagnosis and intervention are early (age < 2 years). This work presents a novel ear-worn wearable EEG system designed to aid in the early detection of ASD. Conventional EEG systems often suffer from bulky, wired electrodes, high power consumption, and a lack of real-time electrode–skin interface (ESI) impedance monitoring. To address these limitations, our system incorporates continuous, long-term EEG recording, on-chip machine learning for real-time ASD prediction, and a passive ESI evaluation system. The passive ESI methodology evaluates impedance using the root mean square voltage of the output signal, considering factors like pressure, electrode surface area, material, gel thickness, and duration. The on-chip machine learning processor, implemented in 180 nm CMOS, occupies a minimal 2.52 mm² of active area while consuming only 0.87 µJ of energy per classification. The performance of this ML processor is validated using the Old Dominion University ASD dataset.

## 1. Introduction

Autism spectrum disorder (ASD) is a complex neurological disorder characterized by a wide range of symptoms and varying degrees of severity. The term “spectrum” reflects this diversity in individual presentations [1]. Children are particularly vulnerable to ASD, with the U.S. Centers for Disease Control and Prevention estimating that 1 in 44 children is affected, and the prevalence continues to rise [2]. Common symptoms of ASD include delayed language development, difficulty with social interaction (such as avoiding eye contact and not responding to their name), repetitive behaviors, and heightened sensitivity to sensory input (like sounds or light) [1].

Diagnosing ASD is a complex process that relies heavily on a comprehensive assessment of developmental history and behavioral observations. Currently, there are no definitive biological markers or medical tests to diagnose ASD [3,4]. The Autism Diagnostic Observation Schedule (ADOS-2) in Figure 1a is a widely used clinical tool for diagnosing ASD. During an ADOS-2 assessment, a clinician evaluates a child’s social communication skills, repetitive behaviors, and imaginative play. By comparing these observations to standardized criteria, the clinician can make a diagnosis of ASD or typical development [4].

Electroencephalography (EEG) is a technique used to record brain activity through electrodes placed on the scalp. Scalp EEG has shown promise in diagnosing various neurological disorders, including epilepsy, Alzheimer’s, and Parkinson’s disease, etc. [1,5,6,7,8,9]. Recent studies have explored the potential of EEG to identify biomarkers for autism spectrum disorder (ASD) [6,7]. Researchers have collected EEG data from individuals with ASD and applied machine learning (ML) and deep learning (DL) algorithms to classify them as either ASD or typically developing (TD).

Wearable biomedical devices have emerged as a promising tool for neurological disorder detection [8,9]. The proposed system in Figure 1b is an ear-worn device that integrates EEG acquisition and electrode–skin impedance (ESI) monitoring to ensure comfort and signal quality. This compact and unobtrusive device enables continuous, 24/7 monitoring of brain activity, allowing for the identification of potential biomarkers for early ASD detection [10]. Early diagnosis is crucial for timely intervention and optimal treatment outcomes, as it has been shown to significantly improve the prognosis for individuals with ASD [11].

The acquisition of high-quality EEG signals from wearable on-chip biomedical processors remains a significant challenge due to various noise sources, including power line interference, motion artifacts, eye blinks, and high electrode–skin impedance (ESI) [8,12]. ESI, in particular, can significantly degrade signal quality [13,14,15] by attenuating the signal, increasing susceptibility to noise, and saturating the low-noise amplifier (LNA). To address these issues, a low-noise analog front end (AFE) capable of acquiring high-quality EEG signals, even in the presence of high ESI [16,17,18], is essential. This work aims to develop an AFE that offers (1) a high signal-to-noise ratio (SNR), (2) a high common-mode rejection ratio (CMRR) [19,20], (3) simultaneous ESI monitoring for quality assurance, and (4) an integrated ASD detection processor. By continuously monitoring ESI, we can ensure that it remains within acceptable limits, leading to improved signal quality and more accurate ASD detection.

The proposed ear-worn EEG acquisition system integrates an on-chip shallow neural network (SNN)-based ASD detection processor, fabricated using a 180 nm CMOS process [21]. The SNN processor utilizes four EEG channels (FP1, FP2, T7, and T8) [22]. While the AFE for high-quality EEG signal acquisition has been designed and tested discretely due to current limitations in acquiring ASD patient data, it is intended to be fully integrated into the system once necessary ethical approvals are obtained.

## 2. Proposed Ear-Worn EEG Acquisition System For ASD Detection

The American Clinical Neurophysiology Society (ACNS) has established guidelines for digital EEG recording and long-term monitoring of neurological disorders [23,24]. These guidelines specify minimum standards for sampling rate (>256 Hz), resolution (≥16-bit), common-mode rejection ratio (>60 dB), input impedance (>1 MΩ), and noise level (<1 μV). Our device adheres to these ACNS standards.

The realized ear-worn EEG recording system comprises an analog front end (AFE) that acquires scalp EEG signals, amplifies them, and removes noise. A back-end microcontroller (BEM) processes the amplified EEG signals, digitizes them, and transmits the data to a digital back-end processor or cloud-based device via Bluetooth Low Energy (BLE). Additionally, the BEM controls the ESI measurement unit and performs impedance calculations. An SNN-based ASD prediction digital back-end (DBE) processor utilizes the digitized EEG data for ASD prediction. Finally, an ESI measurement unit monitors the electrode–skin impedance to ensure signal quality. Figure 2 illustrates the system-level block diagram of the implemented system.

### 2.1. Analog Front End Design

Designing and developing a miniaturized AFE capable of providing low-noise, high-quality EEG signals while ensuring patient comfort is a significant challenge. EEG signals typically range from 1 to 100 microvolts within a 0 to 30 Hz frequency band. The AFE’s first stage comprises a low-noise amplifier (LNA) to amplify the acquired EEG signal and a bandpass filter (BPF) to isolate the desired frequency band. A programmable gain amplifier (PGA) is added as the second stage to further amplify the signal. Additional bandpass and notch filters are included after the PGA to eliminate unwanted frequency components. Figure 3 illustrates the circuit schematics of the proposed AFE. The AFE design is explained in the following section:

#### 2.1.1. Low-Noise Amplifier

The LNA plays a crucial role in amplifying the weak EEG signal while minimizing noise. The LNA’s input stage utilizes an instrumentation amplifier (IA) to amplify the differential analog EEG signal received from the scalp electrodes. This configuration, as depicted in Figure 3, ensures high input impedance, low noise, and excellent common-mode rejection, which are essential for accurate EEG signal acquisition.

We employed the INA333 from Texas Instruments (TI), Dallas, TX, USA as the IA due to its high input impedance, low power consumption, low operating voltage, and high common-mode rejection ratio (CMRR). The INA333, configured with a 24 dB gain, amplifies the differential EEG signal acquired from the scalp electrodes [25]. As depicted in Figure 7 later in this paper, the equivalent impedance model, represented by $E_{HC}$ and $E_{E}$, influences the offset voltage induced when electrodes are placed on the skin. This offset voltage is affected by factors such as electrode type, electrolyte composition, and body temperature [26]. A cascaded low-pass filter (LPF) and high-pass filter (HPF) form a bandpass filter (BPF) with cutoff frequencies of 40 Hz and 0.16 Hz, respectively. This BPF removes any DC components and unwanted frequency elements. The INA333’s high CMRR of over 100 dB significantly reduces the impact of motion artifacts, eye blinks, and power line interference on the differential EEG signal. The proposed AFE offers an input impedance exceeding 1 GΩ and a noise level of approximately 650 nV_*rms*_ within the desired 0.16 Hz to 40 Hz frequency band. These parameters align with the ACNS guidelines for EEG recording [10,23,24].

#### 2.1.2. Programmable Gain Amplifier

A PGA, implemented using the OPA2379 operational amplifier from TI, is employed to further amplify the filtered EEG signals from the LNA. The PGA operates in a non-inverting configuration, providing variable gain. A BPF with a lower cutoff frequency ($f_{L}$) of 0.16 Hz and a higher cutoff frequency ($f_{H}$) of 40 Hz is added after the PGA to isolate the desired frequency band of the EEG signal and eliminate any remaining noise components. To completely suppress the 50 Hz power line interference and electromagnetic interference, a 50 Hz twin-T notch filter (NF) is included after the BPF.

We utilized four EEG channels (FP1, FP2, T7, and T8) to record the EEG data for ASD prediction. The EEG signals from the corresponding AFE were multiplexed using a four-to-one multiplexer and fed into the BEM.

### 2.2. Back-End Micro Controller

The overall power consumption of the device is quite important to ensure the maximum battery life on a single charge. We have therefore chosen the NINA-B400-00B as the BLE module. This ultra-low-power, miniaturized, standalone BLE module from u-blox is powered by the nRF52833 SoC from Nordic Semiconductor. The nRF52833’s integrated RF core and powerful Arm Cortex-M4 processor with floating-point capabilities and ultra-low power consumption were key factors in selecting the NINA-B400-00B as the BEM unit. The maximum dynamic power consumption of the BEM unit is 14.8 mA.

The BEM unit performs two primary functions: (1) digitizing, storing, and transmitting the acquired EEG data; and (2) approximating and transmitting the calculated ESI values for specific durations during EEG data acquisition by the AFE.

#### 2.2.1. Digitization, Storage, and Transmission

The EEG data from the selected four channels (FP1, FP2, T7, and T8) are buffered, multiplexed, and then fed into the built-in 14-bit oversampling successive-approximation-register analog-to-digital converter (SAR-ADC) block of the BEM unit, located at the GPIO25 pin. The ADC digitizes the EEG data at a sampling rate of 256 Hz. The digitized EEG data are stored in the BEM’s memory for subsequent ESI calculations and are transmitted in real time to a mobile phone via Bluetooth Low Energy (BLE). Due to the limited memory capacity of the BEM, the digitized data are also stored on a web server using a mobile application.

#### 2.2.2. Approximate ESI Calculation

The ESI measurement unit employs a switching mechanism involving known resistors (R1 and R2) to determine the electrode–skin impedance. The BEM unit controls this switching process using GPIO pins 4 and 5, generating variable voltage values for different resistor configurations. These voltage values are stored in the BEM for subsequent ESI calculations. The BEM also provides real-time feedback to the user and neurologist regarding the ESI range, alongside the EEG signal. Threshold values of 12 kΩ and 50 kΩ are established for wet pre-gelled Ag/AgCl electrodes and dry electrodes, respectively.

### 2.3. Shallow Neural Network ASD Prediction Processor

An ASD prediction processor is implemented to classify patients as ASD or TPD based on EEG signals. Figure 4 illustrates the block diagram of the ASD prediction processor. The digitized EEG data, either from the AFE or an ASD prediction dataset, is input to the processor. The pre-processing unit then applies down-sampling, mean average referencing, and bandpass filtering to the digitized data to remove noise and artifacts.

The pre-processed EEG data are subjected to feature extraction. For each EEG channel in the beta band, a feature set comprising power spectral density (PSD), Hjorth activity (HJA), and Hjorth mobility (HJM) is computed. The feature set is then normalized using Z-score normalization. The normalized feature set is subsequently fed into a shallow neural network (SNN) for ASD detection. The SNN architecture consists of an input layer, a hidden layer, and an output node with a sigmoid activation function.

#### 2.3.1. ASD Prediction Data Set

An EEG dataset comprising both ASD and TPD individuals is necessary to train the ML processor for ASD detection. In this paper, we utilized the Old Dominion EEG dataset, which provides EEG recordings from fourteen channels for seventeen subjects, including eight with ASD and nine TPD persons [27].

Selecting the minimum number of EEG channels is crucial for developing efficient and cost-effective ASD detection processors. The hardware complexity of the system is directly influenced by the number of channels. In this study, we utilized four EEG channels (FP1, FP2, T7, and T8) for feature extraction [1].

#### 2.3.2. Feature Set and Feature Calculation Unit

The feature calculation unit (FCU) computes three features for each EEG channel in the beta band: PSD, HJA, and HJM. The PSD quantifies the signal energy within the 14 to 32 Hz beta band. HJA represents the variance of the EEG signal in the beta band, while HJM is the square root of the ratio between the signal variance and its first-order derivative within the beta band. Equations (Equation 1)–(Equation 3) outline the calculations for PSD, HJA, and HJM, respectively. Figure 5 illustrates the hardware architecture of the FCU.
(1)$$PSD\left(\beta \right)=\sum_{i=0}^{N}\left[EEG_{i}\right]$$
(2)$$HJA\left(\beta \right)=VAR\left(EEG_{i}\right)$$
(3)$$HJM\left(\beta \right)=\sqrt{\frac{VAR(EEG)}{VAR(EEG^{\prime })}}$$

The bandpassed EEG signal, denoted as $EEG_{i}$ in Equation (Equation 1), is obtained through a 100th order finite impulse response (FIR) filter, as depicted in Figure 5. To calculate the power spectral density (PSD) in the beta band, the bandpassed signal is absolute valued and accumulated. Calculating HJA and HJM traditionally involves complex floating-point operations to compute the variance of the EEG signal and its first-order difference as mentioned in Equations (Equation 2) and (Equation 3). To simplify these calculations, we employ a standard deviation indicator (SDI) based on the range rule to approximate the variance [28]. This approach allows us to avoid floating-point division, leading to the approximated HJA (HJAI) and HJM (HJMI) as defined in Equations (Equation 5) and (Equation 6), respectively.
(4)SDI(β)=MAX(EEG)−MIN(EEG)
(5)$$HJAI\left(\beta \right)=SDI^{2}$$
(6)$$HJMI\left(\beta \right)=SDI\left(EEG\right)-SDI\left(EEG^{\prime }\right)$$

The hardware implementations of HJAI and HJMI are required after approximations of HJA and HJM. The minimum (MIN) and maximum (MAX) values required for SDI in HJAI are calculated after comparing the consecutive EEG samples ($EEG_{i}$ and $EEG_{i-1}$) using a D- flip flop (DFF), as depicted in Figure 5. A comparator compares both values and updates the contents of MIN and MAX using a two-to-one multiplexer. The selection input of the multiplexer is derived by the output of the comparator. The contents of a 32-bit × 2 bits memory block are updated using the outputs of the comparator and multiplexer as index and data, respectively. Similarly, the SDI of the first-order difference of the EEG signal is calculated by subtracting the current EEG sample and the previous EEG sample using a register. The HJAI and HJMI are obtained after squaring the SDI of the EEG and the first-order difference of the EEG. The square root operation in HJM is ignored for HJMI. The proposed FCU implementation avoids complex floating-point operations and does not require huge memory for variance or standard deviation calculations. It calculates the approximated features with ≈64% lower energy. The feature set of twelve features is forwarded to the SNN classifier for classification.

#### 2.3.3. Shallow Neural Network Classification Unit

The SNN for ASD classification has twelve input nodes and an output node. The structure of SNN is shown in Figure 4. The equations for SNN calculation are shown in Equations (Equation 7)–(Equation 10). The hardware architecture for the SNN implementation is shown in Figure 6. The output node is calculated after multiplication and addition of the input features with weights and bias, as in Equation (Equation 7). A sigmoid activation function is applied to calculate the final output of the output node, as in Equation (Equation 8). The sigmoid activation function is defined in Equation (Equation 9).
(7)$$OL_{0}=\sum_{i=0}^{11}F_{i}\cdot W_{i}+B$$
(8)$$OL=Sigmoid(OL_{0})$$
(9)$$Sigmoid\left(X\right)=\frac{1}{1+e^{-x}}$$
(10)Label=(OL≥0.5)

The floating-point multiplier, accumulator, and adder perform the multiplication and accumulation operations for SNN as shown in Figure 6. The $OL_{0}$ from the multiplication and accumulation unit is forwarded to the sigmoid unit for sigmoid function calculation. The sigmoid function defined in Equation (Equation 9) requires complex floating-point operations to calculate $e^{\left(-x\right)}$ and floating-point division operation. To avoid these complex computations, it is implemented using a look-up table. The $OL_{0}$ is single precision floating-point value with 1, 8, and 23 bits for the sign, exponent, and mantissa, respectively.

The $OL_{0}$ is multiplied by a higher number to scale the values above 1. The number is forwarded to the SNN classification unit along with classification parameters. The floating-point multiplication is performed by adding an offset to the exponent part of the $OL_{0}$. For example, the addition of 9 to the exponent multiplies the number by 512. The floating-point-to-integer converter converts the floating-point number to an integer value. The integer index is forwarded to a look-up table of 16 bits × 1024. The look-up table provides a shifted sigmoid value for $OL_{0}$. Since the sigmoid values of positive and negative numbers are symmetrical around 0.5, we have used the shifted sigmoid values, which require 50% lower memory for the look-up table. The actual sigmoid value is obtained after adding or subtracting the shifted sigmoid value from 0.5. The addition or subtraction (negative addition) is performed using a multiplexer with a sign bit as a selection input. The sigmoid value is compared with 0.5 to calculate the SNN label 1 (ASD) or 0 (TPD).

### 2.4. Electrode–Skin Impedance Unit

Pre-gelled Ag/AgCl electrodes or metallic disc electrodes are usually used for skin contact to acquire biopotential signals. The human skin has a complex nature due to which a high-quality EEG signal acquisition is challenging. ESI is the main factor that obstructs the measurement of EEG signals. Different factors including applied pressure [29,30], material type, size, and usage duration of the electrodes affect the ESI measurement. To understand the effect of these factors, Webster’s model [31] is used to depict the ESI during dry and wet (sweat) conditions [32].

Figure 7 depicts the skin layers and corresponding Webster’s equivalent circuit model for the calculation of ESI. $E_{HC}$ is the half-cell potential of the electrode, and $R_{D}$ and $C_{D}$ are the electrode–electrolyte interface resistance and capacitance, respectively. $R_{G}$ is the resistance of the gel, and $E_{E}$ is the potential between the epidermis and the gel. $R_{E}$ and $C_{E}$ are the resistance and capacitance of the epidermis layer, respectively. $R_{S}$ is the resistance of the dermis and subcutaneous tissues of the skin [30,31]. In the presence of human sweat, the ESI model receives an additional RC loop along with the potential added to the dry electrode ESI model in a parallel fashion. $R_{SW}$ and $C_{CW}$ are the resistance and capacitor between the sweat and gel/electrode, respectively. The potential is denoted by the $E_{SW}$. Due to the presence of sweat, the overall impedance changes and results in more low-frequency artifacts [32,33].

Generally, ESI is evaluated by introducing a known current source and measuring the differential voltage across electrodes [17,34]. Conventionally, an active current source is utilized but tends to be power hungry due to current injection in the system [35,36,37]. Hence, the backup time of portable devices decreases significantly [10,34]. The system circuits become complex as they also contain the ESI measuring circuitry. The utilization of active current sources can be risky due to the chance of electric shocks to the person and can also cause an allergy or itching of the skin [38].

To overcome the challenges of the absence of ESI or utilization of active ESI techniques, an ear-worn, multiplexed EEG acquisition device using pre-gelled Ag/AgCl electrodes is implemented [9,39]. The implemented system can measure the real-time ESI intelligently in a closed-loop fashion during the EEG recording. ESI is measured using a novel passive method by attaching known resistors to the input of the LNA [10,38] while observing the changes at the output side. LNA is the first-stage IA that receives the EEG signal from the electrodes connected to the patient’s scalp. LNA should have high input impedance to deal with motion artifacts and fluctuation in ESI [40,41,42]. The placement of input resistances for LNA forms a high pass filter (HPF) having a low cutoff frequency. The cutoff frequency depends on the input resistances [5,42]. The HPF also helps to avoid LNA saturation. The implemented system provides real-time feedback to the person/caregiver about the electrode placement and corresponding ESI values [10,40]. The implemented EEG acquisition system ensures improved quality EEG signals by (a) mitigating the unwanted effects due to ESI, (b) electrode misplacement and other environmental variations that degrade EEG signals. It can also transmit real-time EEG data on the mobile phone and store the data on the webserver to assist the neurologist by providing high-quality EEG recordings. A proposed passive method is applied for the computation of the ESI. No current is injected into the system, therefore no current source is utilized. The voltage level of the EEG signal is determined for the measurement of the ESI. Figure 8 depicts the ESI hardware realization. It consists of resistors (R1 and R2) at the differential input of the LNA having known values of 5.6 kΩ with 0.1% tolerance. The known resistors are connected via switches (IC MAX4741EUA+), controlled by BEM [25]. To compute the ESI, the known resistors are connected in parallel on differential input of the LNA, and the voltage variation of the output signal is measured. The output voltage level is measured in two configurations, i.e., before and after connecting the resistors R1 and R2, respectively. The measured signals are stored in the BEM. The BEM calculates the root mean square voltage (V_*rms*_) and performs the calculation for determining the approximate value of the ESI.

The proposed ESI model with the complete ESI circuit schematic is shown in Figure 8. The Webster model [31] is being utilized for modeling the ESI as indicated in Figure 7. The $R_{S}$, $R_{E}$, and $R_{G}$ are neglected due to their minimal contribution in the final ESI value [35,43]. The EEG bio-signal itself is measured for the calculation of ESI, hence for our EEG-specified limited frequency band of 0.16–40 Hz reactance of capacitors is assumed to be frequency independent and is already incorporated in the equivalent resistances $R_{+}$ and $R_{-}$ of the $Ch1_{+}$ and $Ch1_{-}$ electrodes, respectively. After these assumptions, the equivalent Webster model for ESI becomes a voltage source with a series resistance as shown in Figure 8. $V_{+}$ and $V_{-}$ are the voltage amplitudes of the EEG signal from the brain at the $Ch1_{+}$ and $Ch1_{-}$ electrodes, respectively. The electrode offset voltage is shown by the battery $E_{HC+}$ and $E_{HC-}$ for the positive and negative channel electrodes, respectively. Known value parallel resistors, connected for the measurement of ESI, are represented by R1 and R2. Equation (Equation 11) provides the voltage $V_{+In}$ at the positive input terminal of LNA when $In_{+}$ of LNA is in normal mode ($R_{1}$ is not connected). Equation (Equation 12) provides the voltage $V_{+In}$ at the positive input terminal of LNA when parallel resistor R1 is connected to the positive terminal of LNA. Equation (Equation 13) provides the measured ESI. Similarly, these equations are also applicable to the negative channel. All the processing and calculations for measuring the ESI are performed by the BEM.
(11)$$V_{+In}=V_{+}\times \frac{R_{3}}{R_{3}+X_{C1}+R_{+}}$$
(12)$$V_{+In}=\frac{R_{3}}{R_{3}+X_{C1}}\times \frac{R_{1}}{R_{1}+R_{+}}\times V_{+}$$
(13)$$R_{+}=\frac{R_{1}R_{3}}{R_{3}+X_{C1}}\times \frac{V_{+}}{V_{+In}}-R_{1}$$
(14)$$Z_{+}\approx R_{+}$$

As pre-gelled Ag/AgCl electrodes are used during the measurement procedure, a threshold of 15 kΩ is selected for the impedance status. An impedance value up to 15 KΩ is considered to be normal/low impedance as in the literature impedance around 15 kΩ is acceptable for good quality signal acquisition [35]. If the impedance value is above 15 kΩ, the BEM will update the user that the impedance is high via an indicator in the mobile phone application. After receiving the message on the mobile phone, the user performs the necessary steps including the system’s overall gain adjustment via PGA control and checking the placement of the electrodes as to whether they are properly placed or not.

## 3. Device Power Profile and Prototype

A rechargeable behind-the-ear wearable EEG system is presented. The complete system-level power management and flow profile of the AFE is depicted in Figure 9. A standard type-B micro USB connector on the lateral bottom side of the device is provided which makes it fully operational during charging. An STC 4054 standalone linear Li-Ion battery charger is used to charge the battery. The system is powered up using a 3.7 V 300 mAh Li-ion polymer battery. The system operating voltage is set at 1.8 V in order to reduce the overall power consumption. STLQ020C18R from ST Microelectronics, Geneva, Switzerland. provides a 1.8 V linear voltage regulation with ultra-low quiescent (0.3 μA) current and is responsible for providing the constant 1.8 V to the system. Similarly, the mid-supply reference voltage of 0.9 V is provided by a low-impedance output buffer to ensure the low-impedance voltage supply to the whole system.

The fabricated and assembled printed circuit board (PCB) device is depicted in Figure 10. The microcontroller, PGA control, LNA, and PGA in the PCB device are highlighted in the figure. The circuitry related to the AFE and BEM is on the top side of the PCB while the ESI circuit is on the bottom side of the board. The fabricated PCB and assembled (soldered components) total thicknesses are 0.8 mm and 3 mm, respectively. The arc-shaped PCB is designed to be easily applied behind the ear having X and Y dimensions of only 52 mm × 53 mm, respectively. Figure 10a shows the fabricated PCB of the device. The completely assembled device is shown in Figure 10b. The device enclosure is designed in SOLID-WORKS and developed using a Laser 3D Resin printer. The overall effective (between two extreme points) dimensions of the device’s prototype are 57.5 mm × 29 mm. The PGA gain control is provided on the front side of the device while the charging cable connector slot is provided on the bottom lateral side.

The mobile application’s graphical user interface monitors the acquired EEG signals, ASD prediction, and the measured ESI values using the proposed miniaturized ear-wearable device; along with the feedback regarding electrode placement and ESI measurement, it is shown in Figure 10c. The feedback regarding electrode placement is also provided on the mobile application. An alarm beep for high ESI is enabled. The complete PCB is designed in Altium Designer21 and fabricated from the *JLCPCB* PCB fabrication facility.

## 4. Measurements and Results

The proposed device is tested and validated using the EEG and ESI measurements. The acquired results are compared with the related state-of-the-art works. Table 1 summarizes the comparison of this work with the previous works. This system is the first (to the best of our knowledge), wearable behind-the-ear, ultra-low-power device capable of acquiring an EEG signal along with ESI measurement in line with ACNS Standards [24]. The ESI measurement is obtained using a novel passive technique with minimum power consumption and no shock risks. The previous devices are not rechargeable [5,44,45]. The previous works are also not wearable [44,45], and have a larger area than this work [5]. The previous works also do not provide ESI measurement [5] or use an active ESI measurement methodology [44,45]. The active ESI measurement has a higher risk of shocks or skin abrasions, and therefore is not safe for humans.

### 4.1. Electroencephalogram Measurement

The EEG signal is measured during various states including open, closed, and blinking eyes for testing our designed device for a total duration of 1 min. Figure 11 shows the recorded EEG signals with open, closed, and blinking eyes along with their respective spectrograms. The EEG signals are recorded using wet pre-gelled Ag/AgCl electrodes. The EEG signal recording with open eyes and its spectrogram are shown in Figure 11a and Figure 11b, respectively. The EEG signal recording with closed eyes and its spectrogram are shown in Figure 11c and Figure 11d, respectively. The EEG signal recording with blinking eyes and its spectrogram are shown in Figure 11e and Figure 11f, respectively.

The EEG signal with open eyes with some eye blinking activity for a duration of 5 s was also recorded. Figure 12 shows the recorded EEG signal before and after necessary pre-processing. EEG signals are recorded using dry electrodes. The raw EEG signal and the pre-processed EEG signals are shown in Figure 12a and Figure 12b respectively.

### 4.2. Analog Front End Measurement

The performance of the EEG setup is measured in the lab. Figure 13 depicts the AFE measurement results. Figure 13a shows that the proposed system offers a CMRR of >90 dB approximately in the desired frequency bands. Therefore, a good common noise interference rejection is provided.

The input-referenced noise power spectral density of 0.65 $\mu V_{rms}$ is measured in the frequency band of 0.1–100 Hz using the proposed AFE as shown in Figure 13b.

The AFE variable gains are also measured, Figure 13c indicating an overall gain of >70 dB for the presented AFE.

### 4.3. Electrode Skin Interface Impedance Measurement

#### 4.3.1. Types of Electrodes

Various types of electrodes having different skin contact areas were used for the ESI measurement. Wet electrodes were pre-gelled Ag/AgCl electrodes with a circular metallic sensor disk having diameters of 8 mm and 10 mm, respectively. Dry electrodes made from Ag/AgCl with diameters of 7.2 mm and 10 mm, respectively, were used. The dry electrodes were also tested with skin preparation using Ten20 Conductive Neurodiagnostic Electrode Paste. The details and information on the various electrodes used to measure ESI are shown in Figure 14.

Figure 14a shows the dry reusable Ag/AgCl electrode from PLUX Biosignals with a 7.2 mm diameter metallic disk sensor and a total skin contact area of 40.70 mm ${\,}^{2}$. Figure 14b indicates the disposable wet pre-gelled Ag/AgCl electrode (Kendall ECG Electrodes, H124SG) having 8 mm diameter and 201 mm ${\,}^{2}$ skin contact area, procured from DigiKey (SEN-12969). Figure 14c shows the wet pre-gelled Ag/AgCl electrode with a metallic disk sensor diameter and skin contact area of 10 mm and 314 mm ${\,}^{2}$, respectively. Figure 14d indicates the Ten20 Conductive Neurodiagnostic Electrode Paste by WEAVER and Company for the skin preparation of dry electrodes.

#### 4.3.2. Electrode Skin Interface Impedance Measurement Test

The designed device was used to measure the ESI using various electrodes. The electrodes were tested under two conditions, namely normal state and pressure state. An elastic headband was used for the exertion of pressure. The ESI is a critical parameter for the quality transmission of EEG signals. Low ESI means a good quality EEG signal and vice versa. Various parameters on which ESI is dependent were studied and the results obtained are presented as follows:Effect of Pressure:The effect of the pressure on ESI was observed using both dry reusable Ag/AgCl electrodes and wet disposable Ag/AgCL electrodes. Figure 15 shows the effect of pressure on ESI measurement. Electrodes were placed at the Fp1 and Fp2 locations on the forehead. It was observed that there was no significant effect of pressure on the wet Ag/AgCl electrodes for ESI as depicted in Figure 15A. However, for dry electrodes, a significant decrease in ESI was observed until a specific minimum point was reached at which the electrodes were firmly in contact with the skin [9,46,47,48]. Figure 15B shows the effect of pressure on ESI measurement using dry Ag/AgCl electrodes. A pressure of 15 mm Hg was applied using an elastic headband for trials no 2, 4, 6, 8, and 10. This experiment was repeated five times. This analysis suggests that the electrodes should be in firm contact with the skin to ensure a small ESI value for dry electrodes.Multiple ESI measurements with the designed device were obtained using different dry and pre-gelled Ag/AgCl electrodes of variable diameter (8 mm, 10 mm, and 7.2 mm, 10 mm). Figure 15 shows the ESI measurements of two different sizes of pre-gelled Ag/AgCl electrodes at various time stamps. Time Stamp number 4 shows the ESI when adequate pressure was applied to the measuring electrodes. It can be observed that the ESI decreases when sufficient pressure is applied [29,30] for both wet and dry electrodes as shown in Figure 15A and Figure 15B, respectively.
Figure 15(**A**) Effect of pressure on ESI with the wet disposable pre-gelled Ag/AgCl electrodes and (**B**) dry reusable Ag/AgCl electrodes.
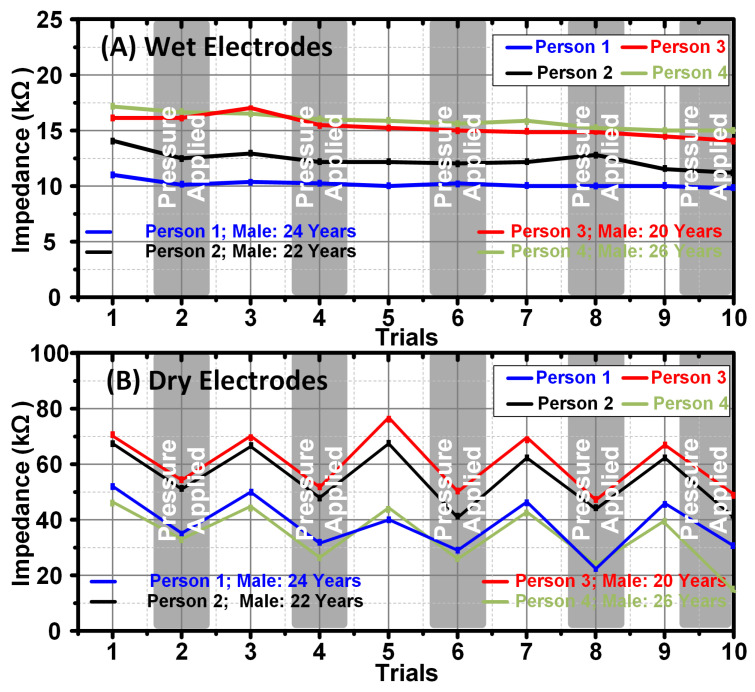

Effect of Skin Preparation on Dry Electrodes:The effect of skin preparation was also observed for the dry reusable Ag/AgCl electrodes. Skin preparation was performed by cleaning the forehead with an alcohol pad and then applying Ten20 conductive electrode paste. Figure 16 shows the effect of skin preparation on the ESI of the dry AG/AgCl electrodes. A slight decrease in ESI for skin preparation as compared to unprepared skin can be observed.
Figure 16Effect of skin preparation on dry Ag/AgCl electrodes for ESI.
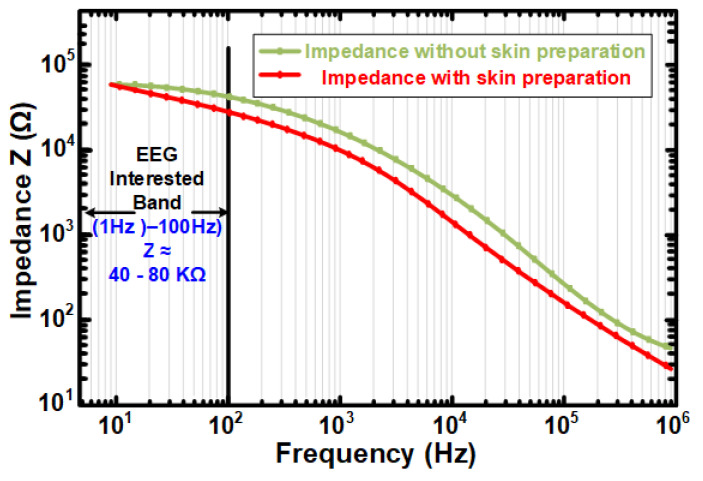

Skin Locations:To verify the design device and validity of ESI measurement, the electrodes were placed at different body locations including the forehead and forearm with a distance of 30 cm between the electrodes. The effect of skin location on the ESI is shown in Figure 17. The nature and composition of the skin at various body locations are different due to stratum corneum thickness, sweat glands, hairs, etc.Maximum ESI was observed on the forearms while minimum ESI was observed on the forehead for both the wet and dry electrodes as shown in Figure 17a. Figure 17b and Figure 17c show the wet electrode ESI measurement and dry electrode ESI measurement setup respectively. Figure 17d shows the dry electrode ESI measurement setup with pressure using an elastic band.
Figure 17(**a**) Effect of skin locations on the electrodes. (**b**) Wet electrode ESI measurement on forehead. (**c**) Dry electrode ESI measurement on forearm. (**d**) Dry electrode ESI measurement under pressure.
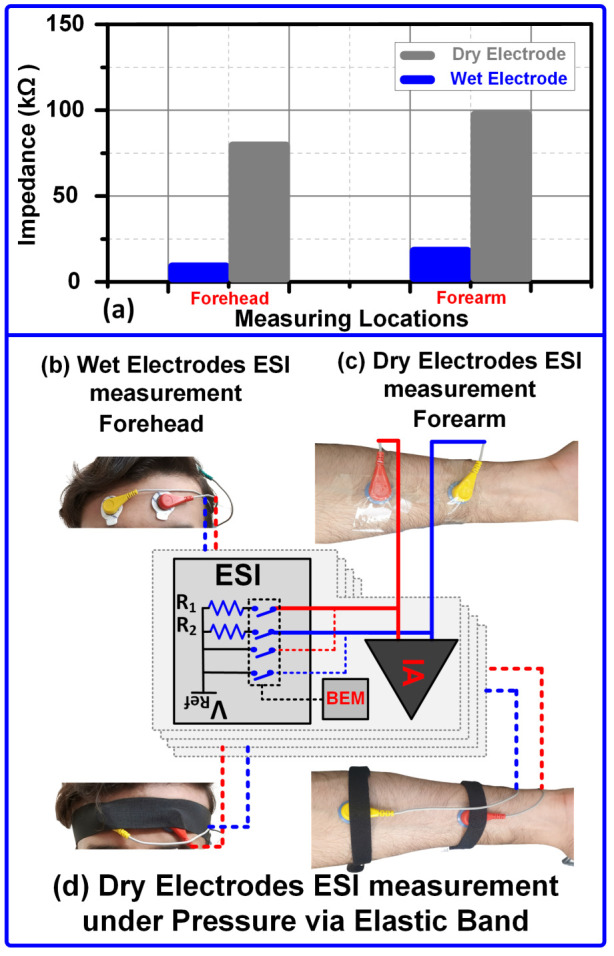


### 4.4. SNN-Based ASD Prediction Processor

Figure 18 shows the SNN ASD detection processor micrograph and its performance summary. The 2.52 mm^2^ processor is implemented in a 0.18 µm CMOS process. It consumes 112 μW to decide between ASD and TPD continuously using digitized EEG data. The implemented processor performs ASD detection with 87.3% accuracy on the ODU data set.

The measurement results for the ASD detection processor are shown in Figure 19. It shows the selected EEG samples (T7 Channel) of two subjects including S2 and S5 in the data set. S2 and S5 are a TPD person and ASD patient, respectively. The normalized features calculated for the S2 and S5 using the selected four channels are forwarded to the SNN unit for detection as ASD or TPD. S2 and S5 are classified as TPD and ASD, respectively.

## 5. Conclusions

This paper introduces a novel, behind-the-ear wearable EEG acquisition device designed to minimize patient discomfort and facilitate extended monitoring periods. The device ensures high-quality EEG signal acquisition by continuously monitoring and maintaining low electrode–skin impedance (ESI). A unique, passive, energy-efficient, and safe technique is employed to measure ESI by connecting known resistors to the LNA input. The acquired EEG data is stored on a server and provided in real time to a mobile phone. This wearable EEG acquisition device is integrated with an on-chip ASD detection processor for autism detection, offering a compact, wireless solution. The device provides real-time feedback to the user regarding EEG signals, electrode placement, and ESI.

## Figures and Tables

**Figure 1 sensors-24-07489-f001:**
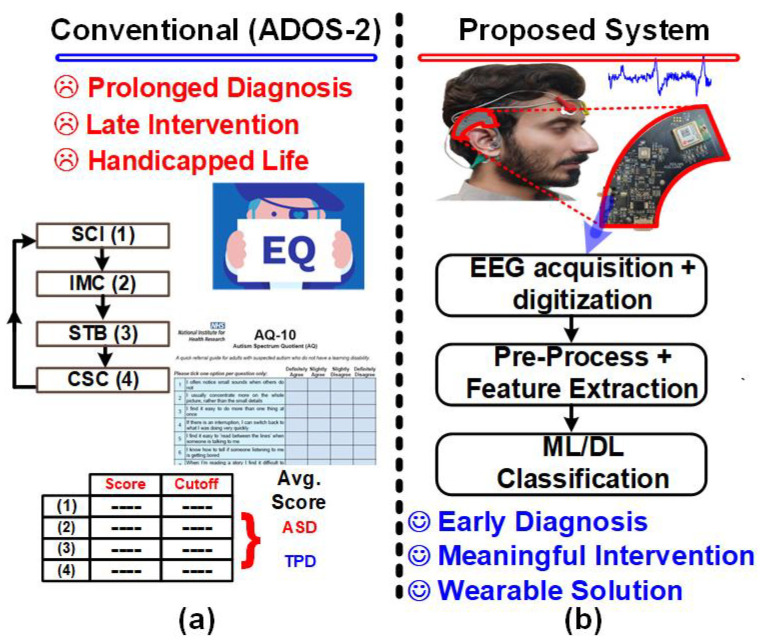
(**a**) Conventional Autism Diagnostic Observation Schedule (ADOS-2) for ASD diagnosis along with (**b**) proposed ASD prediction system.

**Figure 2 sensors-24-07489-f002:**
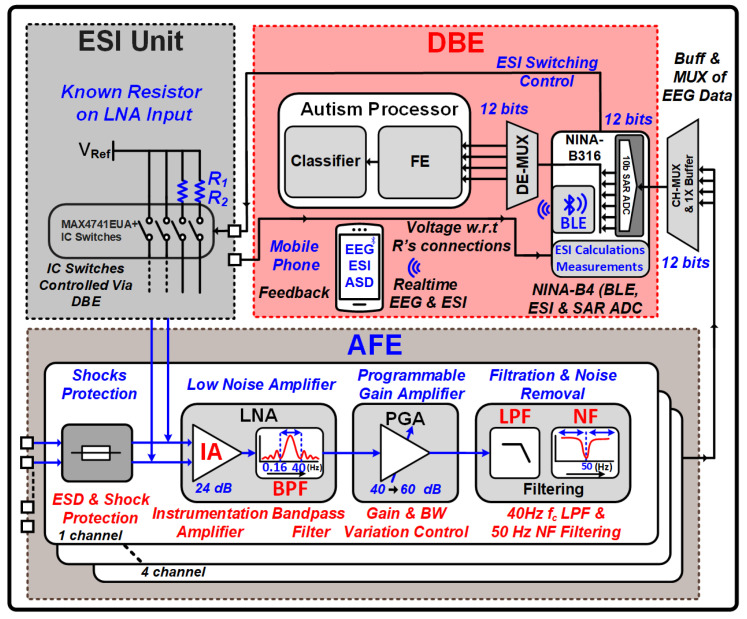
Proposed ear-worn EEG acquisition, ASD prediction, and real-time ESI monitoring system.

**Figure 3 sensors-24-07489-f003:**
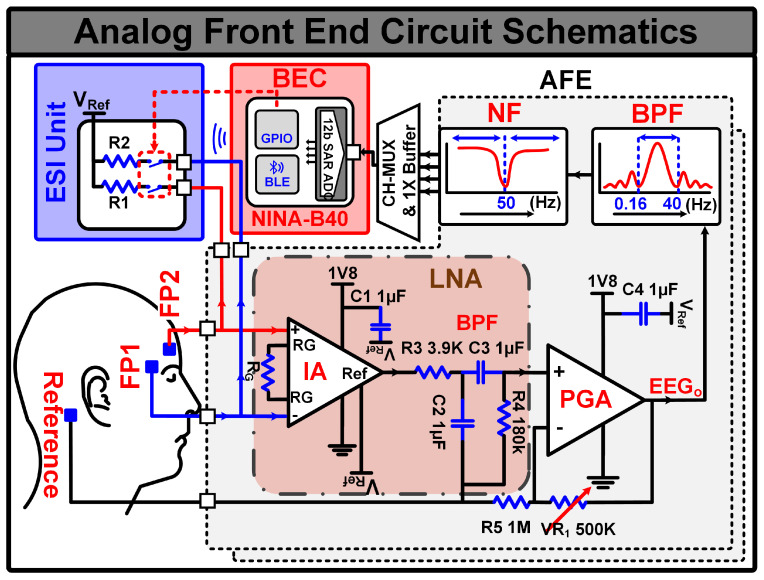
AFE schematic along with electrode placement and ESI circuit.

**Figure 4 sensors-24-07489-f004:**
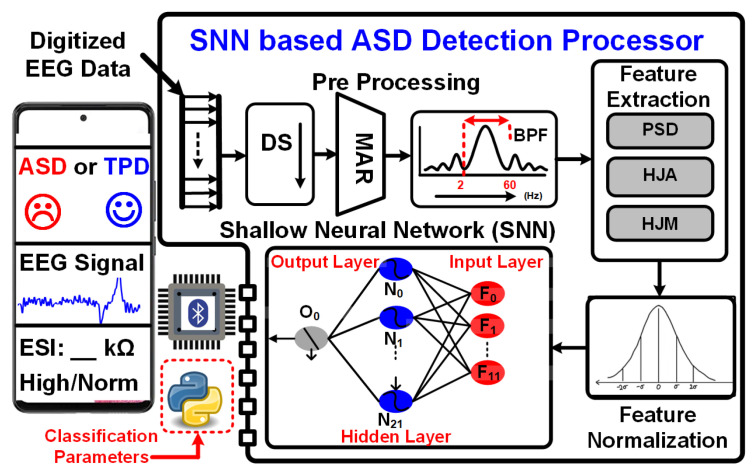
ASD classification processor block diagram.

**Figure 5 sensors-24-07489-f005:**
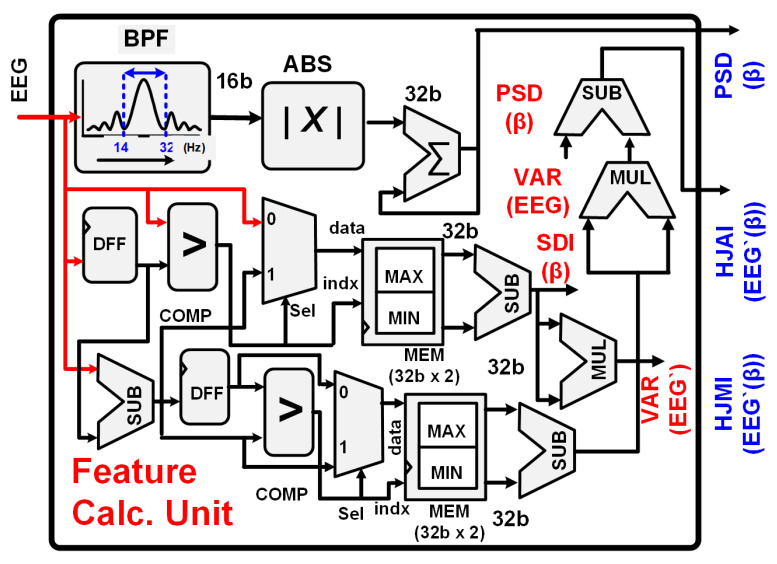
Feature calculation unit’s hardware implementation.

**Figure 6 sensors-24-07489-f006:**
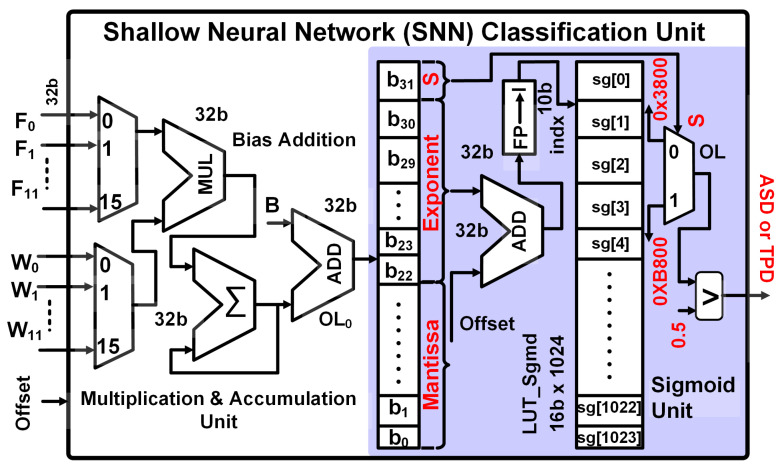
SNN unit’s hardware implementation.

**Figure 7 sensors-24-07489-f007:**
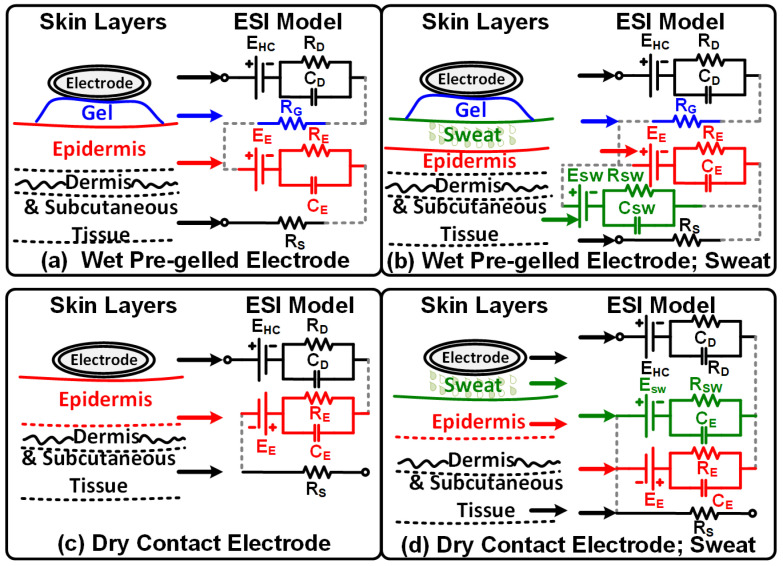
Skin layers and respective equivalent Webster ESI model for wet and dry electrodes under different conditions (no sweating and sweating conditions).

**Figure 8 sensors-24-07489-f008:**
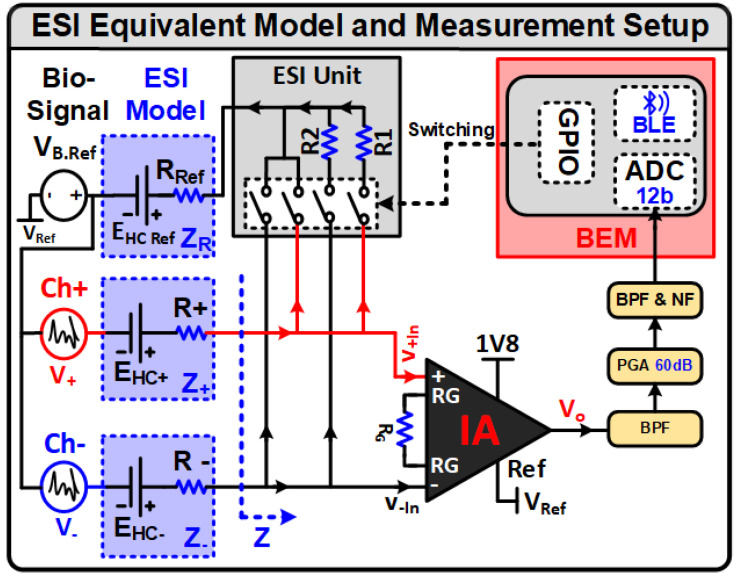
Configuration schematics and measuring model for approximate ESI.

**Figure 9 sensors-24-07489-f009:**
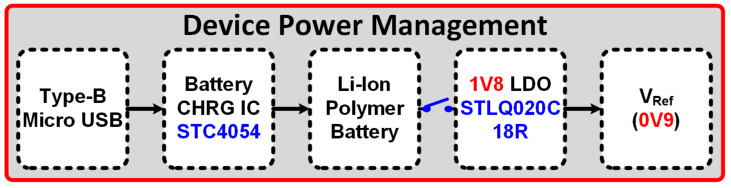
AFE power profile details.

**Figure 10 sensors-24-07489-f010:**
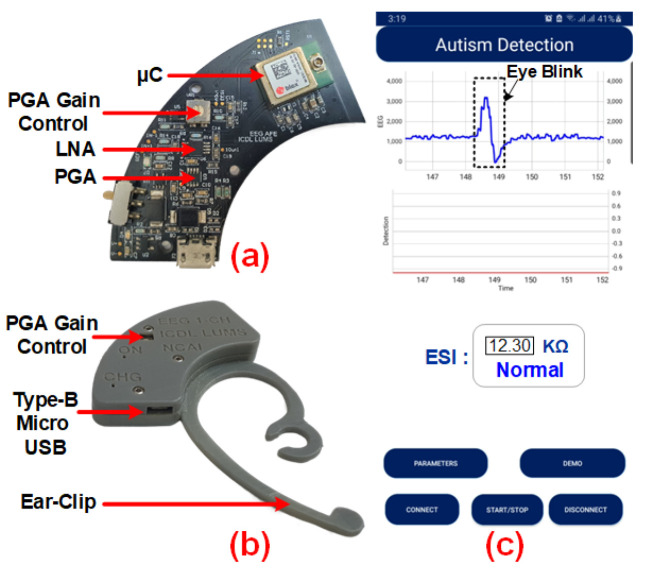
(**a**) Fabricated PCB. (**b**) Completely assembled device. (**c**) Mobile application for real-time EEG, ESI measurement, and ASD classification.

**Figure 11 sensors-24-07489-f011:**
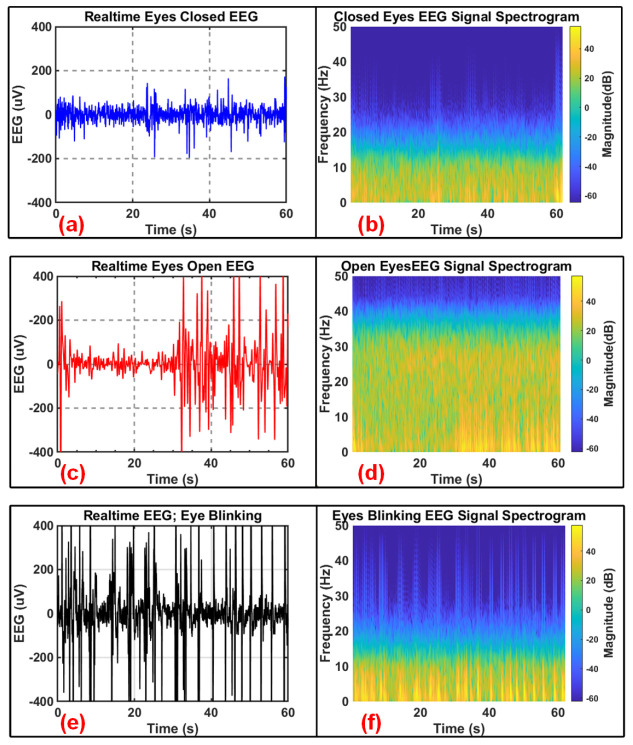
EEG acquisition (Fp1 and Fp2) for various conditions including (**a**) closed eyes, (**c**) open eyes, and (**e**) blinking eyes along with the respective signal spectrograms (**b**,**d**,**f**) for 1 min using wet pre-gelled Ag/AgCl electrodes.

**Figure 12 sensors-24-07489-f012:**
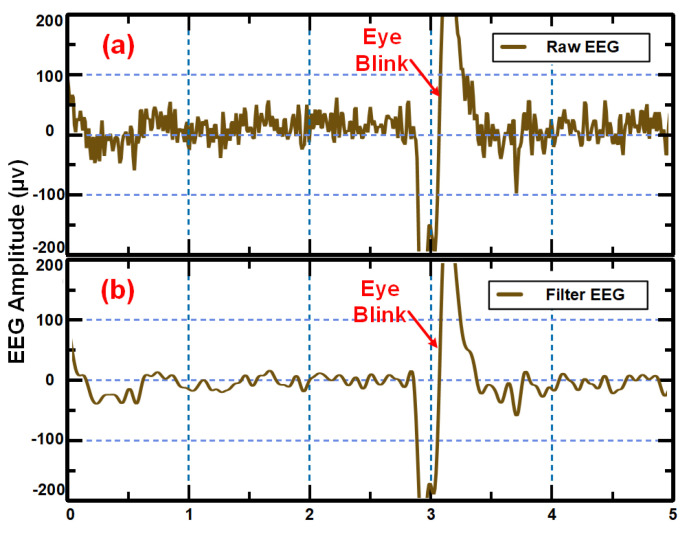
(**a**) Raw and (**b**) filtered EEG signal acquired using the developed device for a window of five seconds using dry electrodes.

**Figure 13 sensors-24-07489-f013:**
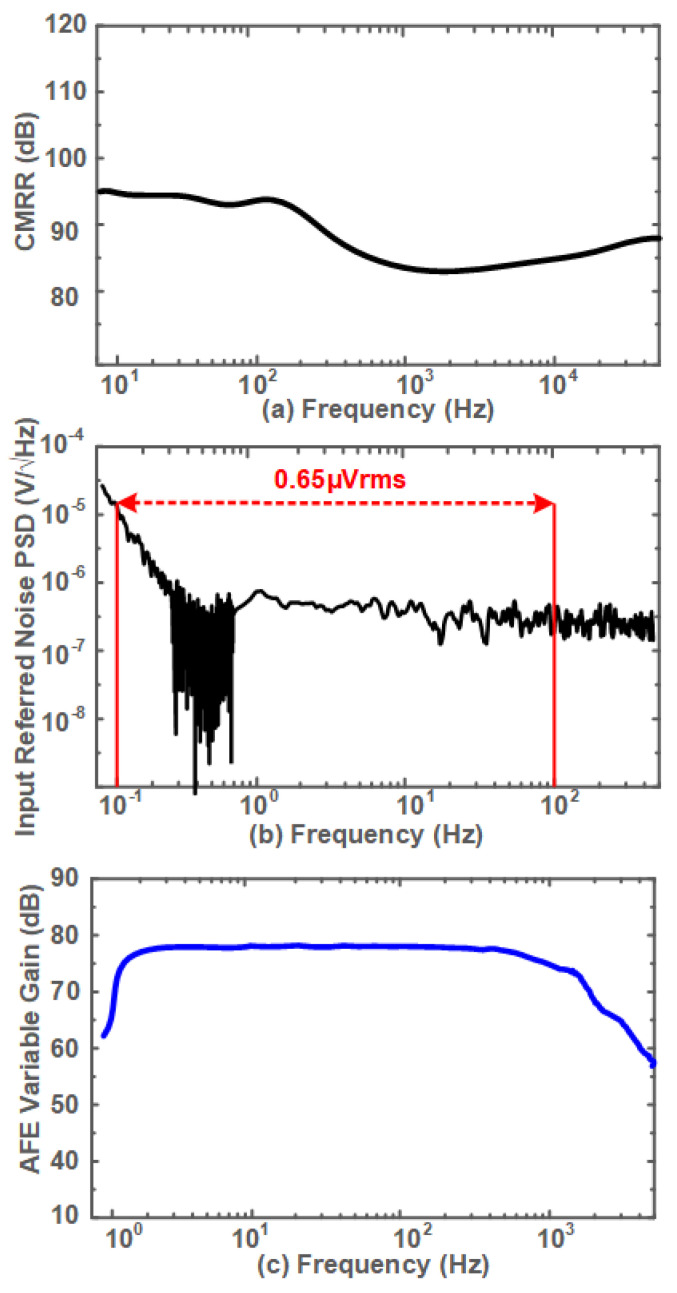
EEG AFE characteristic measurements results for (**a**) CMRR, (**b**) input referred noise, and (**c**) AFE variable gain.

**Figure 14 sensors-24-07489-f014:**
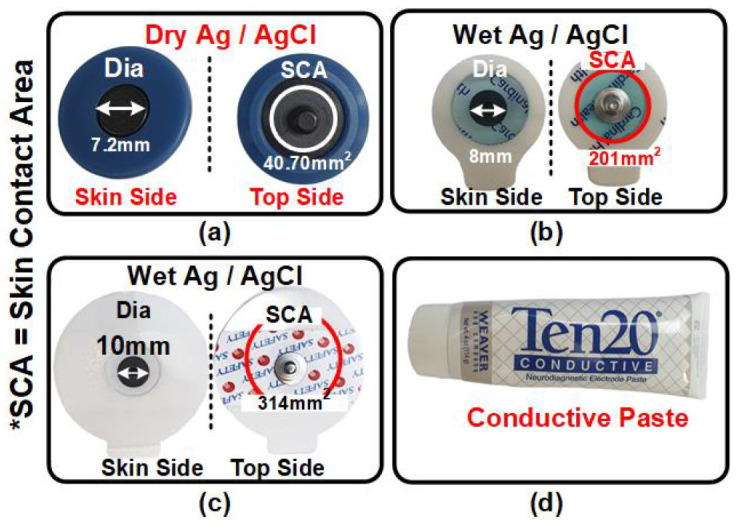
(**a**) Various types of electrodes along with the effective asking contact area (SCA *). The 7.2 mm diameter dry electrodes, (**b**) 8 mm diameter wet electrodes, (**c**) 10 mm diameter wet electrodes, and (**d**) conductive paste.

**Figure 18 sensors-24-07489-f018:**
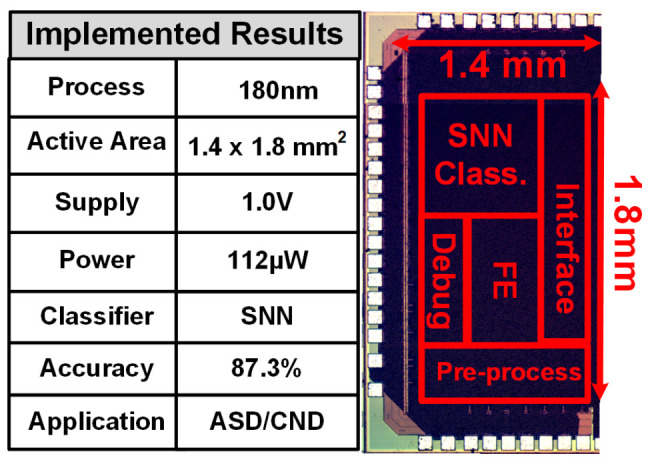
SNN processor die photo and performance summary.

**Figure 19 sensors-24-07489-f019:**
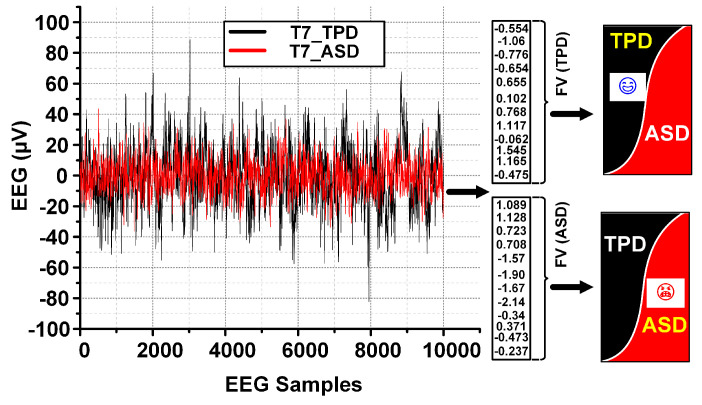
ASD detection processor measurement results.

**Table 1 sensors-24-07489-t001:** State-of-the-art work comparison.

Parameters	EMBC ’14 [5]	Nature ’20 [38]	EMBC ’18 [44]	JOP’16 [45]	This Work
Rechargeable	No	Yes	No	No	Yes
Wearable	Yes	-	No	No	Yes
Dimensions (${\mathrm{mm}}^{2}$)	60 × 64	-	-	-	52 × 53
ESI Measurement	X	(Passive)	(Active)	(Active)	(Passive)
Shocks Risk	No	No	Yes	Yes	No
PGA	No	No	No	No	Yes

## Data Availability

The datasets primarily used for this study was obtained on request from the authors of the following work: Brihadiswaran, G., Haputhanthri, D., Gunathilaka, S., Meedeniya, D., & Jayarathna, S. (2019). EEG-based processing and classification methodologies for autism spectrum disorder: A review. Journal of Computer Science, 15(8), https://thescipub.com/abstract/jcssp.2019.1161.1183, (accessed on 14 November 2024).

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
