# Peer review of "A Closed-Loop Ear-Worn Wearable EEG System with Real-Time Passive Electrode Skin Impedance Measurement for Early Autism Detection†"

_sensors, 2024, doi:10.3390/s24237489_

Round 1

Reviewer 1 Report

Comments and Suggestions for Authors

Dear authors,

I have read carefully your research article.

However, there are a few issues that need to be clarified.

1.- Please double-check the redaction along the paper. There are a few ideas that are repeated once and again that reduce the research quality.

2.- Improve whole the figure captions.

3.- In section 2.1.2, the slew rate for such amplifier might generate some issues?

4.- In section 2.2.1, if you're considering a 14 bits ADC why are you using a 254 Hz frequency? is there any sort of issue to increase the sampling frequency?

5.- In section 2.3.3, there is a flag for the sigmoid function that might generate some issues at -3dB? and can you please elaborate in why do not use a linear function?

6.- In section 2.4, can you list the frequency and the RC at which each of the Fig. 7 circuits are working?

7.- In section 4.1, While acquiring the Fp1 & Fp2, the maximum amplitude for Fig. 7c & e, what is the saturation voltage? as for Fig. 7b, d & f, the central frequency fora each case is?

8.- What is the central frequency that was used to filter it?

9.- What do you require to enhance the detection close to 95%?

Comments on the Quality of English Language

There are plenty of room along the paper to be improved. There are quite a few issues where the authors rewrite the ideas once and again. It does diminish the research quality being performed.

Author Response

Comment 1: Please double-check the redaction along the paper. There are a few ideas that are repeated once and again that reduce the research quality.

Response 1: 

  • First of all, we would like to thank you for sparing your precious time reviewing our manuscript. We are also thankful to you for your valuable comments.
  • We have revised the paper to address your valuable concern so that quality of paper is not reduced. 
  • We once again thank you for this comment.

Comment 2: Improve whole the figure captions.

Response 2: 

  • Thank you for this valuable comment. We have revised the Figure captions.

Comment 3: In section 2.1.2, the slew rate for such amplifier might generate some issues?.

Response 3: 

  • Thank you again for this valuable comment. 
  • Yes, an inappropriate slew rate can cause signal distortion including clipping, harmonic distortion, phase shift etc). It can also reduce bandwidth and affect transient response. To address this, we have chosen an amplifier with the best suitable slew rate. We have also optimized our circuit design with suitable signal conditioning.

Comment 4: In section 2.2.1, if you're considering a 14-bit ADC why are you using a 254 Hz frequency? Is there any sort of issue to increase the sampling frequency?

Response 4: 

  • Thank you again for this valuable comment.  
  • We are using this sampling frequency for our target application of ASD prediction, which uses EEG signals to capture high-frequency components and improve signal fidelity. 
  • Yes, this high frequency can cause some issues including signal noise, data processing requirements, and high power consumption. However, we have addressed these issues by appropriate optimization techniques.

Comment 5: In section 2.3.3, there is a flag for the sigmoid function that might generate some issues at -3dB? and can you please elaborate in why do not use a linear function?

Response 5: 

  • Thank you again for this valuable comment.  
  • Yes, the sigmoid function can cause issues with vanishing gradients and saturated outputs.  However, the linear function could not learn the model for our target applications for ASD prediction.  However, we have minimized these issues by data normalization, batch normalization during model training, and appropriate gradient clipping.

Comment 6: 6.- In section 2.4, can you list the frequency and the RC at which each of the Fig. 7 circuits are working?

Response 6: 

  • Thank you again for this valuable comment.  
  • Since, the main novelty of our work is that we have proposed a passive technique for the ESI measurement. Passive ESI is independent of frequency/alternative signals because it reduces the detailed Webster model (Fig. 7) to a power supply with a resistor R in series as shown in Fig. 8. Connecting known resistor value at the input enables us to estimate the Resistor R, which is controlled by Back End Microcontroller (BEM). The measurements for ESI components were in the range of 10k-15k Ohm for wet pre-gelled electrodes and the 15kOhm threshold was set for ESI to alert users regarding high ESI.
  • We highly appreciate your question and hope that it will answer your concern. 

Comment 7: In section 4.1, While acquiring the Fp1 & Fp2, the maximum amplitude for Fig. 11c & e, what is the saturation voltage? As for Fig. 7b, d & f, the central frequency for each case is?

Response 7:

  • Thank you again for this valuable comment.  
  • The proposed Analog Front End (AFE) has a maximum input dynamics range of 1000uVpp. Hence the proposed system has a saturation of around 1mVpp. In the case of Open eyes state a max amplitude of 800uV was observed and in the case of eyes blinking, amplitude ranges approximately around 950uV. 
  • In the eyes closed state, alpha wave presence was observed and we have a central frequency of 10Hz. In the eyes open situation the beta wave played the action and a central frequency of approx 20Hz is observed. While the eye blinking scenario has a central frequency of approx 5Hz.

Comment 8: What is the central frequency that was used to filter it?

Response 8:

  • Thank you again for this valuable comment.  
  • AFE is a 2-stage amplifier and comprises bandpass filters after each stage and lastly a 50Hz twin-T notch filter. The bandpass filters have high and low cut-off frequencies of 0.16Hz and 40 Hz, respectively. Digital signal processing was applied to remove any leftover noise and unwanted frequency components.
  • Thank you once again for your valuable comment, which reflects your deep interest in reviewing the paper.

Comment 9:
What do you require to enhance the detection close to 95%?

Response 9: 

  • Thank you again for this valuable comment.  
  • To achieve 95% detection accuracy, we need to improve our data quality and increase model complexity, hyperparameter tuning, regularization techniques, and advanced methods like transfer learning. But on the other hand, it would also significantly increase the hardware resources.

    Comment on the Quality of English Language: There are plenty of room along the paper to be improved. There are quite a few issues where the authors rewrite the ideas once and again. It does diminish the research quality being performed.

     Response :

  • Thank you again for this valuable comment.  

  • We have reviewed the whole paper. We have completely rewritten the abstract, introduction, and conclusion sections. We have also revised other sections.

Reviewer 2 Report

Comments and Suggestions for Authors

This paper reported a closed loop ear-worn wearable EEG system with real-time passive electrode skin impedance measurement for early autism detection. Although being interesting, I find that there are some minor issues with the paper that require addressing prior to this being considered for publication in this journal. I have identified the main points for consideration below:

1. This manuscript has some spelling typos, style errors and grammatical errors. Please carefully check the whole manuscript.

2. Some recent references related to electrode-skin impedance for wet, semi-dry and dry electrodes are recommended to be cited, such as SmartMat. 2024;5: e1173; Sensors and Actuators B 241 (2017) 1244–1255; Sensors & Actuators: B. Chemical 277 (2018) 250–260.

3.The electrode-skin impedance is an important index for the interface stability. So, I am curious about how the impedance of ear-worn wearable EEG system changes over time.

4.The signal quality of the ear-worn wearable EEG system should be quantificationally compared with the wet electrodes.

5. The impedance value highly depends on the frequency. So, the frequency for the electrode-skin impedance in Fig. 17(a) should be reported in the revised manuscript.

Author Response

Comments 1: This manuscript has some spelling typos, style errors and grammatical errors. Please carefully check the whole manuscript.

Response 1: 

  • First, we are highly thankful to you for reviewing our manuscript and for your valuable comments.
  • We have reviewed the manuscript and corrected the typos and grammatical errors.

Comments 2: Some recent references related to electrode-skin impedance for wet, semi-dry and dry electrodes are recommended to be cited, such as SmartMat. 2024;5: e1173; Sensors and Actuators B 241 (2017) 1244–1255; Sensors & Actuators: B. Chemical 277 (2018) 250–260.

Response 2: 

  • Thank you once again for your valuable comment. We are thankful to you for recommending these excellent works to be cited. 
  • We have included the mentioned references in the manuscript in Section 4.3.2. The text is also highlighted in red color. 

Comments 3: The electrode-skin impedance is an important index for interface stability. So, I am curious about how the impedance of ear-worn wearable EEG systems changes over time.

Response 3: 

  • Thank you once again for your valuable comment. 
  • Yes, the Electrode-skin impedance change affects EEG signal quality. It can change due to skin condition, electrode design, skin contact, movement, environment and over time. We have addressed this issue by ESI monitoring, signal processing techniques, and providing proper guidelines for the users. Talking about the overall ESI behaviour, initially, immediately after applying the electrodes, high impedance was observed. We conducted the experiments after the settling time of a few minutes and finished the experiment within a few hours to avoid prolonged recordings as drying of electrode gell and paste results in high ESI, less skin contact, and poor EEG quality. So, In the beginning of a few minutes, we observed high ESI, which gradually decreased and remained settled in the standard range for a few hours. Finally, an increase in impedance was observed after a significant time.      

Comments 4: The signal quality of the ear-worn wearable EEG system should be quantificationally compared with the wet electrodes.

Response 4: 

  • Thank you once again for your valuable comment, which reflects your deep interest in reviewing the paper. Direct quantitative comparison with wet electrodes is ideal. The EEG results shown in Fig 11 are acquired using wet electrodes, while Fig 12 depicts the dry electrode results. In the case of dry electrode recordings, high ESI and more proneness to noise were observed. We have updated and added the electrode type details in the corresponding manuscript sections. 
  • We have also provided suitable comparisons w.r.t to ESI, contact area, pressure, and material type for dry and wet electrodes in Fig 15, Fig 16 and Fig 17.

Comments 5: The impedance value highly depends on the frequency. So, the frequency for the electrode-skin impedance in Fig. 17(a) should be reported in the revised manuscript.

Response 5: 

  • Thank you once again for your valuable comment. We proposed a novel passive technique for measuring the ESI, by adding a known value resistor at the input side to approximate the ESI, enabling us to measure ESI using the potential drop in the EEG signal itself upon attaching the resistor without the injection of the active variable current source, detailed explanation in Section 2.4, which is why frequency is not specified in Fig 17 (a). However, we have also done the ESI measurement over the frequency range of a few Hz to 10K Hz, which can be seen in Fig 16, along with passive ESI measurement for the dry electrodes with and without skin preparation.